# Pain Assessment Using the Analgesia Nociception Index (ANI) in Patients Undergoing General Anesthesia: A Systematic Review and Meta-Analysis

**DOI:** 10.3390/jpm13101461

**Published:** 2023-10-04

**Authors:** Min Kyoung Kim, Geun Joo Choi, Kyung Seo Oh, Sang Phil Lee, Hyun Kang

**Affiliations:** 1Department of Anesthesiology and Pain Medicine, College of Medicine, Chung-Ang University, Seoul 06974, Republic of Korea; ddolpaki86@naver.com (M.K.K.); pistis23@naver.com (G.J.C.); oks940418@cauhs.or.kr (K.S.O.); 2Department of Anesthesiology and Pain Medicine, Chung-Ang University Gwangmyeong Hospital, Gwangmyeong-si 14353, Republic of Korea; 3Department of Anesthesiology and Pain Medicine, Chung-Ang University Hospital, Seoul 06973, Republic of Korea; 4Department of Biomedical Engineering Graduate School, Chungbuk National University, Cheongju-si 28644, Republic of Korea; splee@chungbuk.ac.kr

**Keywords:** analgesia, nociception, analgesia nociception index, analgesics, opioid, monitoring, intra-operative

## Abstract

The analgesia nociception index (ANI) has emerged as a potential measurement for objective pain assessment during general anesthesia. This systematic review and meta-analysis aimed to evaluate the accuracy and effectiveness of ANI in assessing intra- and post-operative pain in patients undergoing general anesthesia. We conducted a comprehensive search of Ovid-MEDLINE, Ovid-EMBASE, Cochrane Central Register of Controlled Trials, Google Scholar, public clinical trial databases (ClinicalTrials and Clinical Research Information Service), and OpenSIGLE to identify relevant studies published prior to May 2023 and included studies that evaluated the accuracy and effectiveness of ANI for intra- or post-operative pain assessment during general anesthesia. Among the 962 studies identified, 30 met the eligibility criteria and were included in the systematic review, and 17 were included in the meta-analysis. For predicting intra-operative pain, pooled sensitivity, specificity, diagnostic odds ratio (DOR), and area under curve of ANI were 0.81 (95% confidence interval [CI] = 0.79–0.83; I^2^ = 68.2%), 0.93 (95% CI = 0.92–0.93; I^2^ = 99.8%), 2.32 (95% CI = 1.33–3.30; I^2^ = 61.7%), and 0.77 (95% CI = 0.76–0.78; I^2^ = 87.4%), respectively. ANI values and changes in intra-operative hemodynamic variables showed statistically significant correlations. For predicting post-operative pain, pooled sensitivity, specificity, and DOR of ANI were 0.90 (95% CI = 0.87–0.93; I^2^ = 58.7%), 0.51 (95% CI = 0.49–0.52; I^2^ = 99.9%), and 3.38 (95% CI = 2.87–3.88; I^2^ = 81.2%), respectively. ANI monitoring in patients undergoing surgery under general anesthesia is a valuable measurement for predicting intra- and post-operative pain. It reduces the use of intra-operative opioids and aids in pain management throughout the perioperative period.

## 1. Introduction

Anesthesia required for surgical procedures consists of three interrelated components: hypnosis, analgesia, and muscle relaxation. Balanced anesthesia maximizes effectiveness while minimizing side effects by appropriately adjusting multiple anesthetic agents that target these three components to achieve stability and prevent unwanted autonomic reflexes. Accurate assessment of each component is crucial for optimal anesthesia. In particular, the assessment of nociception and effective pain management are of the utmost importance, as they are closely associated with post-operative pain intensity and complications.

Pain is a subjective experience that involves an intricate process of signal transmission within the autonomic nervous system (ANS) and central nervous system (CNS), spanning the spinal cord, brainstem, and thalamus. Consequently, several pain assessment methods have been developed. In the past two decades, diverse monitoring tools have been devised to evaluate nociception, which refers to physiological responses to noxious stimuli [1,2]. While there are monitoring methods based on the CNS, such as response entropy, and methods based on spinal reflexes, such as the nociceptive flexor reflex, the most commonly employed approach in clinical practice is monitoring based on the ANS.

Owing to the overlapping neuro-anatomy between pain transmission and ANS pathways, pain induces changes in ANS activity, particularly an increase in sympathetic nervous system activation and a decrease in the parasympathetic nervous system [3]. ANS-based pain monitoring observes these changes via surrogate markers of the ANS, including pupillary changes, skin sweating, heart rate variability, pulse wave amplitude, and pulse beat interval. Among these markers, the analgesia nociception index (ANI) is a notable monitoring tool that employs heart rate variability. By analyzing high-frequency adjusted heart rate variability, the ANI quantifies parasympathetic activity on a numerical scale ranging from 0 (indicating maximum pain) to 100 (indicating no pain).

Various studies [4,5,6] have supported the use of the ANI for intra-operative pain monitoring, demonstrating its accuracy and effectiveness. However, conflicting findings have also been reported [7,8]. Therefore, this study aimed to assess the accuracy and effectiveness of ANI by analyzing previous studies that monitored ANI in patients under general anesthesia during surgery.

## 2. Materials and Methods

### 2.1. Protocol and Registration

This systematic review and meta-analysis protocol was developed according to the preferred reporting requirements for systematic reviews and meta-analysis protocols (PRISMA-P) statement [9]. The protocol was drafted and registered with the International Prospective Register of Systematic Reviews (registration number: CRD42021277720; accessible at https://www.crd.york.ac.uk/prospero/display_record.php?ID=CRD42021277720) accessed on 10 October 2021. This study was conducted in accordance with the recommendations of the Cochrane Collaboration [10] and the PRISMA statement [11].

### 2.2. Inclusion and Exclusion Criteria

The research questions, inclusion and exclusion criteria were determined before conducting a systematic review and meta-analysis. We included full reports of randomized controlled trials (RCTs), as well as observational studies, such as cohort studies, cross-sectional studies, case–control studies, and case series, which investigated the three study research questions.

We established three research questions and corresponding PICO-SD (participants, intervention, comparison, outcomes, study design). The PICO-SD information is as follows:

Primary outcome: Can ANI monitoring during surgery predict intra- and post-operative pain in patients receiving general anesthesia?

Participants (P): patients receiving surgery under general anesthesiaIntervention (I): ANI monitoringComparison (C): not applicable (NA)Outcome measurements (O): diagnostic accuracy of intra- or post-operative pain
Q1-1sensitivity, specificity, and diagnostic odds ratio (DOR) of ANI for intra-operative pain stimuliQ1-2area under curve (AUC) of accuracy for intra-operative painQ1-3sensitivity, specificity, and DOR of ANI for post-operative painStudy design (SD): RCTs; cohort, cross-sectional, and case–control studies; and case series

Secondary outcome #1: Do ANI values monitored during surgery in patients under general anesthesia correlate with changes in hemodynamic variables observed during surgery?

Participants (P): patients receiving surgery under general anesthesiaIntervention (I): intra-operative ANI valueComparison (C): intra-operative hemodynamic variables, including systolic blood pressure (SBP), mean arterial pressure (MAP), diastolic blood pressure (DBP), and heart rate (HR)Outcome measurements (O): correlationStudy design (SD): RCTs; cohort, cross-sectional, and case–control studies; and case series

Secondary outcome #2: Can ANI monitoring during surgery decrease the amount of intra-operative opioid used compared to non-use of ANI monitoring in patients undergoing surgery under general anesthesia?

Participants (P): patients receiving surgery under general anesthesiaIntervention (I): ANI monitoringComparison (C): non-use of ANI monitoringOutcome measurements (O): amount of intra-operative opioid useStudy design (SD): RCTs; cohort, cross-sectional, and case–control studies; and case series

Studies with the following features were excluded:Review articles, case reports, letters to the editor, and conference abstracts, as well as animal, preclinical, and all other non-relevant studiesStudies with missing outcome measurements of interest

There were no language limitations or date restrictions in our study.

### 2.3. Information Source and Search Strategy

Two independent investigators (KSO and SPL) searched Ovid-MEDLINE, Ovid-EMBASE, the Cochrane Central Register of Controlled Trials, and Google Scholar from database establishment to September 2021 using search terms. The search terms were developed in consultation with a medical librarian and included a combination of free text, medical subject headings, and EMTREE terms for the “Analgesia nociception index”, “ANI monitor”, and “surgery”. To obtain comprehensive results, we searched ClinicalTrials.gov ”https://clinicaltrials.gov/ (accessed on 30 September 2021” and Clinical research information service “https://cris.nih.go.kr/ (accessed on 30 September 2021)” for ongoing and incomplete clinical trials. We also conducted a search of gray literature using OpenSIGLE. In addition, we searched the reference lists of the original articles to ensure that all available studies were included. The search was updated in May 2023. Reference lists were imported into Endnote software 9.3.3 (Thompson Reuters).

### 2.4. Study Selection

Two investigators (MKK and GJC) independently selected studies. In the first stage of study selection, the two investigators reviewed the titles and abstracts of the identified studies. If a study was considered eligible based on the title or abstract in the first stage, the full paper was retrieved and evaluated in the second stage of study selection. We retrieved potentially relevant studies that were identified by at least one investigator or abstracts that did not provide sufficient information regarding the eligibility criteria. The full-text versions of these studies were evaluated. Both investigators discussed their opinions and arrived at a consensus on whether a study should be included or excluded. If an agreement was not reached, the dispute was resolved by a third investigator (HK).

For both stages of study selection, kappa statistics were used to measure the degree of agreement for study selection between the two independent investigators. Kappa statistics were interpreted as follows: (1) less than 0, less than chance agreement; (2) 0.01 to 0.20, slight agreement; (3) 0.21 to 0.40, fair agreement; (4) 0.41 to 0.60, moderate agreement; (5) 0.61 to 0.80, substantial agreement; and (6) 0.8 to 0.99, almost perfect agreement [12]. To minimize data duplication due to multiple reports, articles with the same author, organization, or country were double-checked and excluded.

### 2.5. Data Extraction

To extract all the data necessary for evaluation, a standardized extraction form determined in advance via discussion was used. Two investigators (MKK and GJC) independently extracted the data using the predetermined extraction form. The following information was extracted: (1) title, (2) name of the first author, (3) name of the journal, (4) year of publication, (5) country, (6) language, (7) number of subjects, (8) information for study quality assessment, (9) inclusion criteria, (10) exclusion criteria, (11) sex, (12) age, (13) study protocol registration (registry and registration number) and (14) nature of primary and secondary outcomes.

If the information provided was insufficient or inadequate, the investigators contacted the authors to request additional information. In cases where it was not possible to obtain the required information, missing data were either calculated from the available data or extracted from the figure using the open-source software Plot Digitizer (version 2.6.8); “http://plotdigitizer.sourceforge.net/ (accessed on 20 October 2021)”.

After data extraction, the forms were cross-checked to verify the accuracy and consistency of the extracted data. If any discrepancies were identified, a third investigator (HK) reviewed the data and made the final decisions.

### 2.6. Study Quality Assessment

Two independent investigators (MKK and GJC) critically appraised the quality of each study using tools specific to their study design [13]. The Revised Cochrane Risk of Bias tool for randomized trials (RoB 2.0) was used for RCTs [14], the Risk of Bias Assessment Tool for Nonrandomized Studies (RoBANS) for non-randomized study [15], and the Quality of Diagnostic Accuracy Studies (QUADAS)-2 tool for diagnostic accuracy studies [16]. Investigators rated the following domains: (1) bias arising from the randomization process, (2) bias due to deviations from the intended interventions, (3) bias due to missing outcome data, (4) bias in measurement of the outcome, and (5) bias in selection of the reported result for RoB 2.0; (1) selection of participants, (2) confounding variables, (3) intervention (exposure) measurement, (4) blinding of outcome assessment, (5) incomplete outcome data, and (6) selective outcome reporting for RoBANS; (1) patient selection for the study, (2) reporting index tests, (3) reference content criteria, and (4) flow and timing for QUADAS-2. The methodological quality of the domains in each study was rated as “low risk”, “high risk”, or “unclear”.

For RoB 2.0, the overall risk of bias was determined based on the ratings for each domain. A study was categorized as low risk if all domains were low risk, high risk if at least one domain was high risk or multiple domains had concerns, and some concern if the overall judgment was neither low nor high. Disagreements were resolved via discussions with a third investigator (HK).

### 2.7. Statistical Analysis

Ad hoc tables were designed to summarize the data from the included studies and to show their key characteristics and important questions related to the aim of this review. After data extraction, the reviewers determined whether a meta-analysis was possible. A meta-analysis was conducted when it was feasible to combine the data, whereas in cases where the data provided were unsuitable or lacked sufficient information for synthesis, only a systematic review was conducted.

To evaluate diagnostic performance, we calculated pooled sensitivity and specificity with corresponding 95% confidence intervals (CIs) using the Mantel–Haenszel method of the random-effect model and pooled DOR using the DerSimonian–Laird random-effect model [17]. The DOR can be calculated as the ratio of the odds of positivity in a diseased state to the odds of positivity in a non-diseased state. The value of the DOR ranges from zero to infinity, with higher values indicating a better discriminative performance. A value of 1 indicates that the test does not discriminate between people with and without the disease [18]. We also plotted summary receiver operating characteristic (SROC) curves (using proportional hazards and bivariate models) and estimated the AUC of accuracy. The closer the value of the AUC is to 1, the better the validated diagnostic test [19,20].

The standardized mean difference (SMD) and 95% CIs were calculated for intra-operative opioid use. We used the chi-square test and I^2^ test to explore the heterogeneity between studies. We graded percentages of around 25% (I^2^ 25), 50% (I^2^ 50), and 75% (I^2^ 75) as mild, moderate, and severe heterogeneity, respectively. As P_chi_^2^ was less than 0.10 and the I^2^ value was greater than 50%, the DerSimonian–Laird random-effects model was used [21]. To explore the heterogeneity, we performed a subgroup analysis according to the type of opioid used. Publication bias was not estimated because fewer than 10 studies were included.

Meta-analysis was conducted using Comprehensive Meta-Analysis version 2.0 (Biostat Inc., Tampa, FL, USA), STATA 17.0 (STATA, College Station), and R program (mada package; R Foundation for Statistical Computing).

### 2.8. Quality of the Evidence

The evidence grade for each outcome was evaluated by using the Grading of Recommendations, Assessment, Development, and Evaluation (GRADE) systems. Two investigators (KSO and SPL) performed this process using a sequential assessment of evidence quality, an assessment of the risk-benefit balance, and a subsequent judgment on the strength of the recommendations [22]. Discrepancies were resolved by a third investigator (HK).

## 3. Results

### 3.1. Study Selection

A total of 926 studies were identified via searches of Ovid-MEDLINE, Ovid-EMBASE, the Cochrane Central Register of Controlled Trials, Google Scholar, ClinicalTrials.gov (accessed on 30 September 2021), Clinical research information service (accessed on 30 September 2021), and OpenSIGLE. An additional 36 studies were identified via manual searches. After excluding 134 duplicates from the combined 962 studies, the titles and abstracts of 828 studies were screened in the first selection process. Among these, 784 were excluded based on the screening process. At this stage of study selection, the kappa value for selecting studies between the two reviewers was 0.845. Upon assessment of the full texts of the remaining 44 studies, 14 studies were deemed ineligible for inclusion because of their lack of outcomes of interest [23,24,25,26] and their classification as literature reviews [27,28], letters [29,30], editorial comments [31], conference abstracts [32,33,34], or study protocols [35,36] (Appendix B). The kappa value for the articles selected by the two investigators was 0.891. Consequently, 30 studies met the predefined inclusion criteria. After excluding 13 studies for which the data provided were unsuitable or lacked sufficient information for synthesis, 17 were included in the meta-analysis (Figure 1).

### 3.2. Study Characteristics

The characteristics of the 30 studies in accordance with the rigorous inclusion criteria are described in Table 1. Of the 30 studies, 20 were observational studies [4,6,7,37,38,39,40,41,42,43,44,45,46,47,48,49,50,51,52,53], including one case–control study [53]; nine were RCTs [5,8,54,55,56,57,58,59,60]; and one was a secondary analysis study [61] based on previous research [57]. Three studies [6,50,51] specifically focused on pediatric patients.

Upon examining the specific topics of the studies, we found that 18 studies [4,6,7,37,38,39,40,41,42,44,46,47,48,49,50,51,52,57] investigated the ability of ANI monitoring to predict pain during or after surgery. Additionally, 4 studies [43,45,56,61] examined correlation between hemodynamic changes during surgery and ANI monitoring. Furthermore, 8 studies [5,8,53,54,55,58,59,60] explored the effect of ANI monitoring on intra-operative opioid consumption.

### 3.3. Study Quality Assessment

Table 2 describes the quality assessment performed in this study. For QUADAS-2, all studies were classified as low risk in all domains except the index test. In these studies, index test domains were classified as of some concern because the threshold of the ANI was not prespecified. For ROB 2.0, six studies showed a high risk of bias and three studies showed some concerns.

### 3.4. Quality of the Evidence

Table 3 summarizes the main results and key information regarding the certainty of the evidence evaluated using GRADE systems. The level of current evidence was rated as “low” or “very low”. The major concerns were derived from the inconsistency and imprecision in the prediction of intra-operative and post-operative pain, risk of bias, and inconsistency in intra-operative use of opioid.

### 3.5. Prediction of Intra-operative or Post-operative Pain

#### 3.5.1. Prediction of Intra-operative Pain

Thirteen studies [4,6,38,39,41,42,44,47,48,49,51,52,57] assessed the accuracy of pain reflection in ANI monitoring during general anesthesia. Among these, seven studies [6,41,42,47,49,51,52] used nociceptive stimuli (intubation, surgical incision, tetanic stimulation, and intracutaneous stimulation) as pain criteria, whereas four studies [38,39,44,57] focused on hemodynamic changes as pain criteria. One study [4] observed both criteria. Additionally, one study [48] considered the signal of inadequate anesthesia as a pain criterion.

Of the eight studies that used nociceptive stimuli as pain criteria, three studies (n = 114) [4,6,52] that reported the sensitivity and specificity for painful stimuli were included in the meta-analysis (GRADE evidence: very low). The pooled results suggested that the ANI was accurate at predicting painful intra-operative stimuli, with significant results in pooled sensitivity, specificity, and DOR of ANI (0.81, 95% CI = 0.79–0.83, I^2^ = 68.2%, P_chi_^2^ = 0.043; 0.93, 95% CI = 0.92–0.93, I^2^ = 99.8%, P_chi_^2^ < 0.001; and 2.32, 95% CI = 1.33–3.30, I^2^ = 61.7%, P_chi_^2^ = 0.073, respectively) (Figure 2A–C and Appendix A).

Seven studies reported the AUC of accuracy in predicting intra-operative pain, which was detected using hemodynamic reactivity [4,38,39,44,57], painful stimuli [6] and inadequate anesthesia signals [48]. We excluded Theerth et al. [57] and Jeanne et al. [44] studies from meta-analysis as they only reported the AUC but did not report information on measures of dispersion (such as variance, standard deviation, or CI).

The pooled AUC of ANI in predicting the intra-operative pain was 0.77 (95% CI = 0.76–0.78, I^2^ = 87.4%, P_chi_^2^ < 0.001) (Figure 3A). A sensitivity analysis was performed to rule out the effect of Park et al.’s [48] study results, in which the signal of inadequate analgesia was used as a pain criterion. Excluding the study by Park et al., AUC remained constant, but heterogeneity was slightly increased (0.77, 95% CI = 0.76–0.78, I^2^ = 88.0%, P_chi_^2^ < 0.001) (Figure 3B). Theerth et al. [57] reported an AUC of 0.588–0.687 according to the type of hemodynamics (MBP or HR) or type of ANI (analgesia nociception index instantaneous, ANIi, or analgesia nociception index mean, ANIm) (Table 4).

Other studies that were not included in the meta-analysis also indicated that the ANI reflects the degree of pain or analgesia (Table 3). Some studies have compared changes in ANI values with nociceptive stimulation at different remifentanil infusion rates. In these studies, ANI values decreased after stimulation compared with pre-stimulation, whereas ANI values increased as the infusion rate of remifentanil increased [42,49,51]. The validation of ANI as a pain assessment tool has been further reinforced by other studies in which ANI values decreased following painful stimuli. Jeanne et al. [44] demonstrated a statistically insignificant decrease in ANI following a tibial cut (median, 80.5 to 62; not significant *p* = 0.5) in patients without hemodynamic reactivity, whereas other studies have shown significant decreases in ANI after intubation [41,42,47], surgical incision [41,47], and tetanic stimulation [42] (median or mean range, 33 to 55.7), compared with pre-stimuli (median or mean range, 46.8 to 73.9).

#### 3.5.2. Prediction of Post-Operative Pain

Five studies [7,37,40,46,50] evaluated the applicability of ANI monitoring for predicting post-operative pain levels (GRADE evidence: very low). Four studies (n = 807) were included in the meta-analysis. The pooled results suggested that the ANI was accurate at predicting post-operative pain, with significant results in pooled sensitivity, specificity, and DOR of ANI (0.90, 95% CI = 0.87–0.93, I^2^ = 58.7%, P_chi_^2^ = 0.064; 0.51, 95% CI = 0.49–0.52, I^2^ = 99.9%, P_chi_^2^ < 0.001; and 3.38, 95% CI = 2.87–3.88, I^2^ = 81.2%, P_chi_^2^ = 0.001, respectively) (Figure 4A–C and Appendix A). Sensitivity analysis, excluding one study that showed very low specificity [40], resulted in increased specificity and decreased heterogeneity (0.83, 95% CI = 0.81–0.85, I^2^ = 0.0%, P_chi_^2^ = 0.599) (Appendix A).

One study [46], excluded from the meta-analysis, categorized the pain levels of patients who underwent laparoscopic cholecystectomy into three groups (group I: numerical rating scale (NRS) ≤ 3, group II: NRS of 4–6, and group III: NRS ≥ 7). This study examined the relationship between NRS and ANI in these groups. The findings of this study indicated that the use of ANI as a predictor of post-operative pain was ineffective. In group I, the correlation was minimal (r = 0.016); in group II, there was a negative correlation (r = −0.286), and in group III, there was a positive correlation (r = 0.293). However, another study demonstrated a negative linear relationship between ANI and immediate post-operative pain measured within 10 min of arrival in the recovery room [7]. Furthermore, a study conducted on children in the post-anesthetic care unit (PACU) showed lower ANI scores in patients experiencing moderate to severe pain (Face, Legs, Activity, Cry, Consolability (FLACC) Scale > 4) than those with mild pain (FLACC scale ≤ 4), and indicated an accurate prediction of post-operative pain (AUC of ANIm: 0.94 and AUC of ANIi: 0.83) [50].

### 3.6. Correlation between Hemodynamic Changes and ANI Monitoring

Four studies [43,45,56,61] investigated the correlation between ANI values and hemodynamic variables measured during surgery under general anesthesia (Table 5). The correlation coefficient observed between ANI and heart rate ranged from −0.161 to −0.594, ANI and MAP ranged from −0.091 to −0.534, and ANI and SBP was −0.348. Except for the study conducted by Theerth et al. [56], which reported an insignificant correlation between the mean ANI value and MAP (r = −0.091, *p* = 0.085), all other studies had statistically significant results.

### 3.7. Intra-operative Opioid Consumption and ANI Monitoring

Eight studies (255 patients in control group and 261 patients in opioid group) investigated the association between intra-operative opioid use and ANI monitoring (GRADE evidence: low). Among these studies, two [5,58] used fentanyl, one [8] used morphine, three [55,59,60] used remifentanil, and two [53,54] used sufentanil. One unmatched case–control study [53] was not included in the meta-analysis, and seven studies [5,8,54,55,58,59,60] with 480 patients were.

Intra-operative opioid use was significantly lower in the ANI group than in the non-ANI group (SMD = −0.262, 95% CI = −0.450 to −0.075, I^2^ = 79.24, P_chi_^2^ < 0.001). Subgroup analysis showed reduced intra-operative opioid use in the ANI group when remifentanil was used (SMD = −0.871, 95% CI = −1.179 to −0.564, I^2^ = 0.0, P_chi_^2^ = 0.597), but not when fentanyl, morphine, or sufentanil was used. (SMD = −0.078, 95% CI = −0.452 to 0.296; I^2^ = 0.0, P_chi_^2^ = 0.874; SMD = 0.069, 95% CI = −0.290 to 0.429; SMD: 0.600; 95% CI = 0.021 to 1.178, respectively) (Figure 5). However, Le Gall et al.’s study [53], which was not included in the meta-analysis, demonstrated a statistically significant reduction in sufentanil use in the ANI group (Table 4).

## 4. Discussion

This systematic review and meta-analysis of 30 studies demonstrated that ANI monitoring during surgery can assist in predicting pain in patients during and after the surgical procedure. It also correlates with hemodynamic variables and a reduction in the use of intra-operative opioids during surgery. ANI monitoring, which has no additional risks apart from electrode attachment risk, yielded positive results. These findings contribute to maximizing the benefits of clinical application of ANI.

In clinical practice, adjustments to the administration of narcotic analgesics, inhalation agents, and intravenous anesthetics during general anesthesia are commonly made based on the clinician’s judgment with the aim of stabilizing vital signs. This is achieved by observing the patient’s physiological reaction to nociception, such as hemodynamic changes, sweating, or movement. For instance, if blood pressure or heart rate increases or the patient moves, physicians often attribute this to intra-operative pain and increase the dosage of opioids administered. However, it is important to note that clinical scenarios are not always dichotomous, and a significant number of patients with elevated blood pressure do not necessarily have inadequate analgesia and may not require additional analgesics. Logier et al. [62] reported that in cases of low ANI, arterial hypertension responded to a sufentanil bolus, whereas in cases of high ANI, arterial hypertension responded to nicardipine but not sufentanil. ANI monitoring can be useful for identifying the underlying causes of painless hypertension. Furthermore, it provides an additional advantage of preventing unnecessary opioid use that may occur based solely on hemodynamic monitoring.

One of the methods of assessing the validity of ANI is to compare its correlation with that of hemodynamic monitoring. ANI values, which reflect intra-operative pain in this study, have been found to be significantly correlated with hemodynamic variables. The majority of studies [43,45,61] in this area have consistently demonstrated a significant correlation (correlation coefficient ranging from −0.258 to −0.594), with only a few exceptions [56]. However, it is important to note that changes in hemodynamic variables can be influenced not only by intra-operative pain but also by various factors such as age, intravascular volume status, drugs affecting the ANS, and anesthesia depth and type [63,64]. These factors contribute to the inter-individual variability in hemodynamic responses during surgical procedures. Therefore, hemodynamic reactivity may not be an ideal criterion for evaluating analgesia and nociception levels or for validating ANI values as indicators of pain levels.

A more effective criterion for validating ANI could involve a comparative analysis of ANI changes in response to nociceptive stimuli. ANI changes in the presence or absence of stimuli at various opioid concentrations and following the administration of analgesic agents, such as opioid boluses. Various studies exploring different nociceptive stimuli have demonstrated that ANI values reflect the responses to these stimuli [4,6,52]. ANI values significantly decreased during nociceptive periods compared to non-nociceptive periods, providing evidence that ANI is capable of detecting nociception. Moreover, several studies [42,49,51] examining ANI changes at different opioid infusion rates have shown significant variations in responses to pain stimuli.

For instance, Susano et al. [49] reported that ANI values decreased during tetanic stimulation and increased with increasing infusion concentrations of remifentanil. They also observed that as the concentration of remifentanil increased, the difference in ANI values before and after tetanic stimulation became statistically insignificant. Additionally, Gruenewald et al. [42], with similar protocols, demonstrated comparable results for tetanic stimulation and varying infusion rates of remifentanil. Ledowski et al. [47] reported a significant increase in mean ANI values from 53 ± 17 to 59 ± 19 (mean ± standard deviation) before fentanyl bolus administration to after administration (*p* < 0.05). Similarly, Kommula et al. [45] found a significant increase in the mean ANI values following the administration of a fentanyl bolus. Although a quantitative analysis was not performed in this systematic review and meta-analysis, these findings provide evidence that ANI reflects the balance between nociceptive stimulation and analgesic levels, validating the use of ANI as a monitoring tool for nociception.

This study also found that ANI monitoring reduced the intra-operative opioid use. However, the results varied depending on the type of opioid used. These differences may be attributable to variations in opioid pharmacokinetics. For example, remifentanil has a rapid onset and a short duration of action, making it relatively insensitive in the context of administration. In contrast, opioids, such as morphine or sufentanil, have a slower onset and longer duration of action, thus posing challenges in terms of titration and potentially resulting in significant fluctuations in the concentration of the drug at the site of action [65]. The pharmacokinetic characteristics of opioids appear to influence their titration and overall quantity used in response to changes in ANI values.

This systematic review and meta-analysis also showed ANI exhibited good performance for predicting post-operative pain. Although not included in the quantitative analysis, studies have indicated that ANI values not only reflect intra-operative nociception during surgery, but also post-operative pain. For example, one study [37] found a statistically significant negative linear relationship (r^2^ = 0.33, *p* < 0.01) between ANI levels immediately before extubation and pain scores using the NRS upon arrival in the PACU. Another study [7] showed a similar result (r^2^ = 0.41, *p* < 0.05) between ANI and NRS scores upon arrival at PACU.

However, pain scores reported by patients are highly subjective, and ANI values are affected not only by the pain itself but also by the level of alertness and psychological stress of conscious patients, such as those after surgery [35]. Therefore, the correlation between nociception and ANI values in the intra-operative state may not be equally applicable to awake patients after surgery. Some studies have reported no correlation between ANI and post-operative visual analogue scale [44] or NRS [57] scores. Additionally, studies by Charier et al. [40] and KÖPRÜLÜ et al. [46] showed a weak negative correlation. Although individual studies may show varying results, the pooled results from the meta-analysis in this systematic review and meta-analysis demonstrated that ANI exhibited excellent performance in predicting post-operative pain. Hence, while ANI monitoring can assist in predicting post-operative pain and facilitate the development of post-operative analgesic plans, it is important to acknowledge that the interpretation of ANI values in awake patients may be influenced by multiple variables. Therefore, medical professionals must exercise their clinical judgment by considering these multiple factors.

This study had several limitations. First, after a comprehensive and sensitive search, only 30 studies were included. Some outcomes may have been underpowered; therefore, the findings of this study are inconclusive. Second, in most studies included for predicting intra-operative or post-operative pain, the threshold of ANI was not prespecified, which may have led to overly positive or negative estimates of sensitivity and specificity. Third, many studies we included were retrospective rather than prospective, which leads to a higher risk of bias in patient selection and index test domains. Fourth, considerable heterogeneity was observed in some outcomes. This may be due to variations in the types and severity of surgeries, differences in anesthesia protocols, variations in the use of analgesics, and differences in dosages among the included studies. Fifth, this review only included patients who underwent general anesthesia, as the review aimed to evaluate the validity of ANI for a specific question. Therefore, the validity of these results may differ between patients who are sedated and awake.

## 5. Conclusions

ANI monitoring during surgery can assist in predicting patient pain both during and after the surgical procedure. It also correlated with changes in hemodynamic variables during surgery and a reduction in the use of intra-operative opioids.

## Figures and Tables

**Figure 1 jpm-13-01461-f001:**
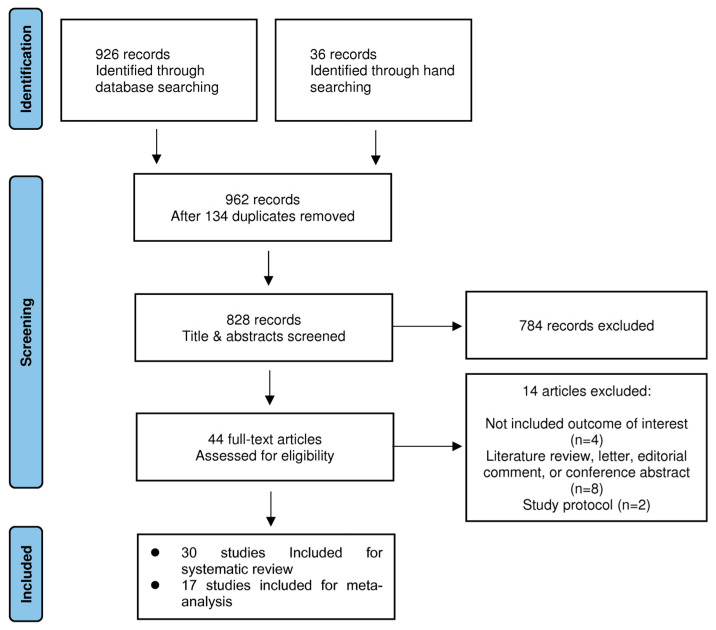
PRISMA flowchart of included and excluded trials.

**Figure 2 jpm-13-01461-f002:**
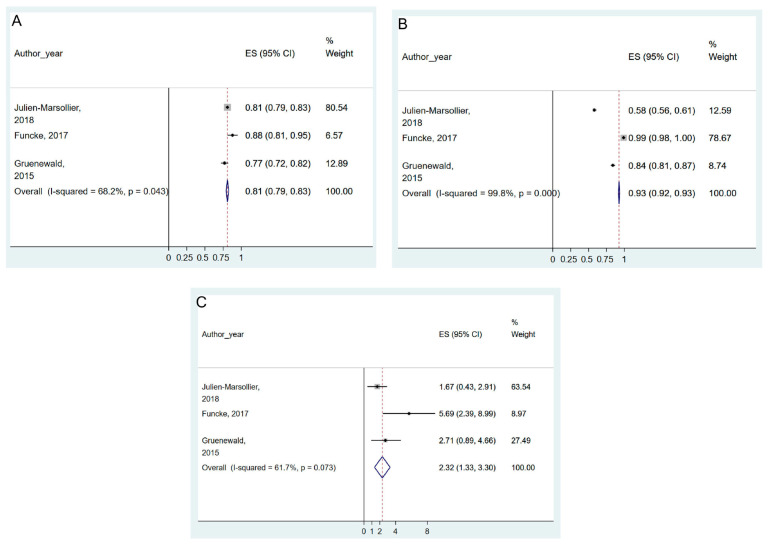
Forest plot of sensitivity (**A**), specificity (**B**), and diagnostic odds ratio (**C**) of ANI for the prediction of intra-operative painful stimuli. The figure depicts individual trials as filled squares with relative weight and the 95% confidence interval (CI) of the difference as a solid line. The diamond shape indicates the pooled estimate and uncertainty for the combined effect. Julien-Marsollier et al., 2018 [6]; Funcke et al., 2017 [52]; Gruenewald et al., 2015 [4].

**Figure 3 jpm-13-01461-f003:**
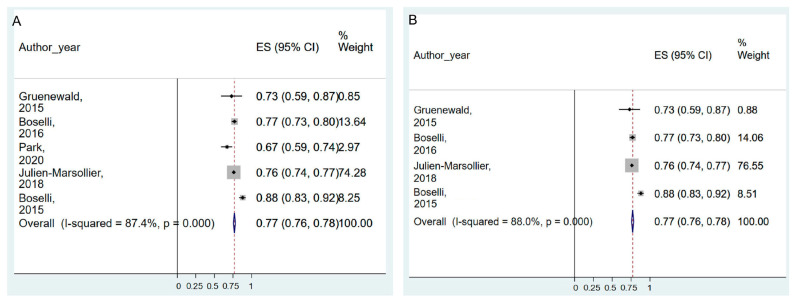
Forest plot of area under curve of accuracy (AUC) in predicting intra-operative pain. (**A**) shows the pooled AUC of ANI for predicting intraoperative pain, while (**B**) presents AUC values excluding the study by Park et al. The figure depicts individual trials as filled squares with relative weight and the 95% confidence interval (CI) of the difference as a solid line. The diamond shape indicates the pooled estimate and uncertainty for the combined effect. Gruenewald et al., 2015 [4]; Boselli et al., 2016 [39]; Park et al., 2020 [48]; Julien-Marsollier et al., 2018 [6]; Boselli et al., 2015 [38].

**Figure 4 jpm-13-01461-f004:**
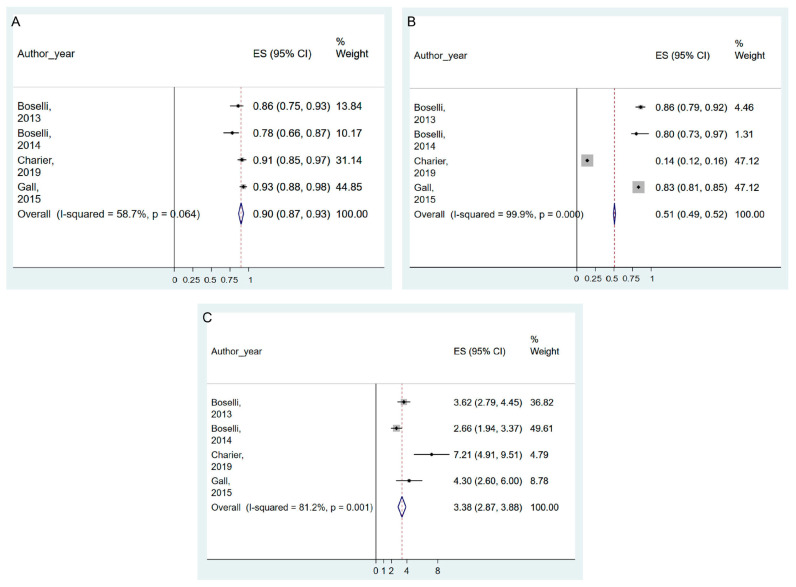
Forest plot of sensitivity (**A**), specificity (**B**), and diagnostic odds ratio (**C**) of ANI for the prediction of post-operative pain. The figure depicts individual trials as filled squares with relative weight and the 95% confidence interval (CI) of the difference as a solid line. The diamond shape indicates the pooled estimate and uncertainty for the combined effect. Boselli et al., 2013 [7]; Boselli et al., 2014 [37]; Charier et al., 2019 [40]; Gall et al., 2015 [50].

**Figure 5 jpm-13-01461-f005:**
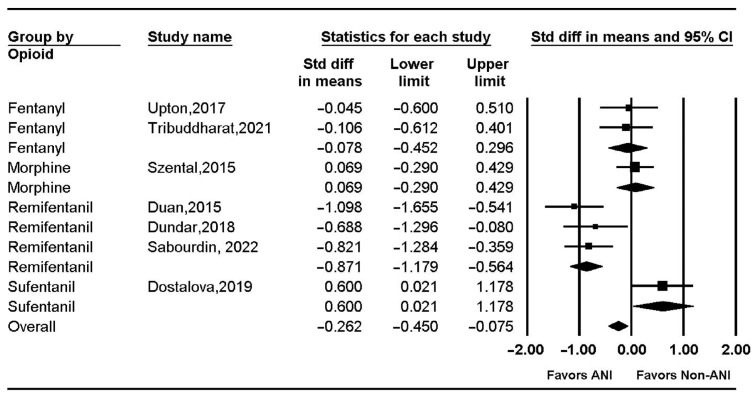
Intra-operative opioid use. The figure depicts individual trials as filled squares with relative weight and the 95% confidence interval (CI) of the difference as a solid line. The diamond shape indicates the pooled estimate and uncertainty for the combined effect. Upton et al., 2017 [5]; Tribuddharat et al., 2021 [58]; Szental et al., 2015 [8]; Duan et al., 2015 [59]; Dundar et al., 2018 [55]; Sabourdin et al., 2022 [60]; Dostalova et al., 2019 [54].

**Table 1 jpm-13-01461-t001:** Characteristics of included studies.

Author (Year), Country	Study Design	Characteristics of Participants (Number)	Anesthetic Agents	Primary Outcome or Purpose
Julien-Marsollier (2018) France [6]	Observational study	ENT, abdominal, urological, or orthopedic surgery involving incision or laparoscopic trocar placement (n = 49)	Propofol Sevoflurane Remifentanil Atracurium	Assessing the diagnostic value of monitoring the ANI to detect surgical stimulation in children
Funcke (2017) Turkey [52]	Observational study	Open radical prostatectomy (n = 37)	Propofol Remifentanil	Correlation between ANI and remifentanil dose administered
Gruenewald (2015) Germany [4]	Observational study	Elective surgery (n = 27)	Sevoflurane Remifentanil Rocuronium	Comparing variations of ANI for noxious stimulations (laryngeal mask airway insertion, tetanic stimulation, intubation) according to remifentanil dose administered
Boselli (2016) France [39]	Observational study	ENT or lower limb orthopedic surgery (n = 128)	Propofol Ketamine Desflurane Remifentanil Cisatracurium	Comparing dynamic variations of ANI and static values to predict hemodynamic reactivity
Park (2020) Korea [48]	Observational study	Stomach surgery, colorectal surgery, hepatobiliary surgery (n = 81)	Propofol Remifentanil Rocuronium	Evaluating the performance of NPI in comparison with the SPI and the ANI in patients under general anesthesia with target-controlled infusion of propofol and remifentanil
Theerth (2018) India [57]	RCT single blind parallel assignment	Brain tumor operation (n = 57) Scalp block (n = 29) Incision infiltration (n = 28)	Thiopentone Sevoflurane Fentanyl Vecuronium	Evaluating intra-operative fentanyl consumption
Susano (2021) Portugal [49]	Observational study	Elective craniotomy (n = 16)	Propofol Remifentanil Rocuronium	Evaluating ability of the ANI monitor to detect standard noxious stimulus (tetanic stimulation) in patients under total intravenous anesthesia with propofol and remifentanil
Coulombe (2021) Canada [41]	Observational study	Elective abdominal surgery via laparotomy (n = 30)	Propofol Desflurane Remifentanil Rocuronium	Documenting ANI variations after standard nociceptive stimulation at 0, 20, and 50% of inhaled N_2_O
KÖPRÜLÜ (2020) Turkey [46]	Observational study	Laparoscopic cholecystectomy (n = 36) NRS ≤ 3 group (n = 17) NRS 4–6 group (n = 11) NRS ≥ 7 group (n = 8)	Midazolam Propofol Sevoflurane Fentanyl Remifentanil Rocuronium	Determining whether or not a correlation exists between the ANI values recorded at the completion of an operation and immediately before and after extubation and the NRS values recorded in the PACU in a group of patients who underwent laparoscopic cholecystectomy, with the goal of evaluating the potential use of ANI values for the prediction of post-operative pain levels
Sabourdin (2013) France [51]	Observational study Pilot study	Middle-ear surgery (n = 12)	Propofol Sevoflurane Desflurane Remifentanil	Describing the profiles of ANI after a standardized nociceptive stimulation, in anesthetized children, at different infusion rates of remifentanil
Ledowski (2014) Australia [47]	Observational study Pilot study	Elective surgery (n = 30)	Propofol Sevoflurane Fentanyl Rocuronium	Determining whether changes to the ANI would coincide with or precede observed haemodynamic changes
Gruenewald (2013) Germany [42]	Observational study	Elective surgery (n = 25)	Propofol Remifentanil	Challenging the ability of ANI to detect standardized noxious stimulation.
Boselli (2013) France [7]	Observation study	ENT, plastic surgery (n = 200)	Ketamine Propofol Sevoflurane Desflurane Remifentanil	Using ANI in the assessment of immediate post-operative pain in adult patients undergoing general anesthesia.
Boselli (2014) France [37]	Observation study	ENT, lower extremity surgery (n = 200)	Ketamine Propofol Sevoflurane Desflurane Remifentanil Cisatracurium	Performing ANI measurements at arousal from general anesthesia to predict immediate post-operative pain on arrival in PACU.
Charier (2019) France [40]	Observational study	Elective surgery (n = 345)	Intravenous anesthetic agents Opioid Volatile anesthetics	Comparing the respective values of ANI, PD, PLR, and VCPD with post-operative VAS scores
Gall (2015) France [50]	Observational study Pilot study	Elective surgery (n = 32) Imaging procedure (n = 30)	No restriction on the anesthetic technique	Investigating the relationship between the ANI and objective measurements of pain intensity during the recovery phase after procedures under general anesthesia in children
Sriganesh (2019) India [61]	RCT secondary analysis of Theerth (2018)	Brain tumor surgery (n = 57) Scalp block (n = 29) Incision infiltration (n = 28)	Thiopentone Sevoflurane Fentanyl Vecuronium	Observing ANI changes during direct laryngoscopy and tracheal intubation
Theerth (2019) India [56]	RCT single blind parallel assignment	Brain tumor operation (n = 57) Scalp block (n = 29) Pin-site infiltration (n = 28)	Thiopentone Sevoflurane Fentanyl Vecuronium	Comparing ANI and hemodynamic changes during skull pin insertion
Jeanne (2012) France [43]	Observational study	Laparoscopic appendectomy or cholecystectomy (n = 15)	Propofol Sevoflurane Remifentanil Cisatracurium	Comparing ANI with heart rate and systolic blood pressure during various noxious stimuli
Kommula (2019) India [45]	Observational study	Craniotomy (n = 21)	Propofol Sevoflurane Fentanyl Vecuronium	Monitoring analgesia during craniotomy using ANI monitor and comparing it with cardiovascular parameters
Boselli (2015) France [38]	Observation study	Suspension laryngoscopy (n = 50)	Propofol Remifentanil	Evaluating the performance of ANI to predict hemodynamic reactivity during suspension laryngoscopy.
Jeanne (2014) France [44]	Observational study	Total knee replacement (n = 27)	Propofol Sufentanil	Determining (1) whether ANI variations could detect early HemodR during propofol anesthesia for total knee replacement, (2) whether ANI measures are coherent with pain after surgery when patients are in PACU, and (3) the threshold predictive of HemodR to prepare for an interventional clinical trial that would measure the benefit of using the ANI monitor to adapt opioids during general anesthesia
Tribuddharat (2021) Thailand [58]	RCT double blind parallel assignment	Mastectomy ANI group (n = 30) Anesthesiologist’s judgment (n = 30)	Propofol Desflurane Fentanyl Cisatracurium	Comparing the efficacy of ANI with standard pharmacokinetic pattern to guide intra-operative fentanyl administration.
Upton (2017) Australia [5]	RCT single blind parallel assignment	Discectomy/laminectomy ANI (n = 24) Anesthesiologist’s judgment group (n = 26)	Propofol Sevoflurane Fentanyl Rocuronium	Documenting post-operative NRS pain score from 0 to 90 min of PACU stay Investigating the effect of intra-operative ANI-guided fentanyl administration on post-operative pain
Szental (2015) Australia [8]	RCT single blind parallel assignment	Laparoscopic cholecystectomy ANI group (n = 59) Anesthesiologist’s judgment group (n = 60)	Propofol Sevoflurane Desflurane Fentanyl Morphine Neuromuscular blocking agent (anesthetists’ choice)	Assessing post-operative moderate/severe pain (VAS ≥ 50 mm) at any of the four time points in the first post-operative hour
Dundar (2018) Turkey [55]	RCT single blind parallel assignment	Breast surgery ANI group (n = 22) Anesthesiologist’s judgment group (n = 22)	Propofol Sevoflurane Fentanyl Rocuronium	Measuring total intra-operative remifentanil consumption Evaluating the effectiveness of ANI monitoring during intra-operative period
Duan (2015) China [59]	RCT single blind parallel assignment	Posterior lumbar spine surgery ANI group (n = 28) Anesthesiologist’s judgment (n = 29)	Propofol Remifentanil Vecuronium	Investigating the effect of intra-operative ANI-guided remifentanil administration
Sabourdin (2022) France [60]	RCT single blind parallel assignment	Gynecologic surgery ANI group (n = 38) Anesthesiologist’s judgment group (n = 40)	Propofol Remifentanil Atracurium	Measuring ntra-operative remifentanil consumption
Dostalova (2019) Czech Republic [54]	RCT single blind parallel assignment	Spine surgery ANI group (n = 24) SPI group (n = 24) Anesthesiologist’s judgment group (n = 24)	Propofol Desflurane Sufentanil Atracurium	Measuring total intra-operative dose of sufentanil Comparing patterns of intra-operative use of opioids, post-operative cortisol levels and post-operative pain scores
Le Gall (2019) France [53]	Unmatched case control study	Bariatric surgery ANI group (n = 30) from prospective cohort Anesthesiologist’s judgment (n = 30) from retrospective cohort	Propofol Sevoflurane Sufentanil Succinylcholine	Measuring mean hourly intra-operative sufentanil requirement Comparing intra-operative opioid consumption

Anesthesiologist’s judgment group; the intra-operative opioid infusion rate was regulated at the anesthesiologist’s discretion or a pre-established protocol when there were signs of inadequate analgesia, such as tears, pupil dilation, sweating, tachycardia, or hypertension. ANI, analgesia nociception index; ENT, ear–nose–throat; HemodR, hemodynamic reactivity; NPI, nasal photoplethysmographic index; NRS, numerical rating scale; PACU, post-anesthesia care unit; PD, pupillary diameter; PLR, pupillary light reflex; SPI, surgical plethysmograhic index; VAS, visual analogue scale; VCPD, variation coefficient of pupillary diameter.

**Table 2 jpm-13-01461-t002:** Risk of bias assessment.

Quadas-2						
Study	Domain					
	Patient Selection		Index Test	Reference Standard	Flow and Timing	
Julien-Marsollier (2018) [6]	Low		Some concerns	Low	Low	
Funcke (2017) [52]	Low		Some concerns	Low	Low	
Gruenewald (2015) [4]	Low		Some concerns	Low	Low	
Boselli (2013) [7]	Low		Some concerns	Low	Low	
Boselli (2014) [37]	Low		Some concerns	Low	Low	
Charier (2019) [40]	Low		Some concerns	Low	Low	
Gall (2015) [50]	Low		Some concerns	Low	Low	
**RoBANS**						
**Study**	**Domain**					
	**Selection of Participants**	**Confounding Variables**	**Intervention** **Measurement**	**Blinding of** **Outcome Assessment**	**Incomplete** **Outcome Data**	**Selective Outcome Reporting**
Julien-Marsollier (2018) [6]	Unclear	No	No	Yes	No	No
Funcke (2017) [52]	Unclear	No	No	Yes	Yes	No
Gruenewald (2015) [4]	Unclear	No	No	Yes	Yes	No
Boselli (2016) [39]	Unclear	No	No	Yes	Yes	No
Park (2020) [48]	Unclear	No	No	Yes	Yes	No
Susano (2021) [49]	No	No	No	Yes	No	No
Coulombe (2021) [41]	Unclear	No	No	Yes	No	No
KÖPRÜLÜ (2020) [46]	Unclear	No	No	Yes	No	No
Sabourdin (2013) [51]	Unclear	No	No	Yes	No	No
Ledowski (2014) [47]	Unclear	No	No	Yes	No	No
Gruenewald (2013) [42]	Unclear	No	No	Yes	Yes	No
Boselli (2013) [7]	Unclear	No	No	Yes	No	No
Boselli (2014) [37]	Unclear	No	No	Yes	No	No
Charier (2019) [40]	Unclear	No	No	Yes	No	No
Gall (2015) [50]	Unclear	No	No	Yes	No	No
Jeanne (2012) [43]	Unclear	No	No	Yes	No	No
Kommula (2019) [45]	Unclear	No	No	Yes	No	No
Boselli (2015) [38]	Unclear	No	No	No	No	No
Jeanne (2014) [44]	Unclear	No	No	No	No	No
Le Gall (2019) [53]	Unclear	No	No	Yes	No	No
Sriganesh (2019) [61]	Unclear	No	No	No	No	No
**RoB 2.0**						
**Study**	**Domain**					
	**Randomization Process**	**Deviations from Intended Interventions**	**Missing Outcome Data**	**Measurement of** **the Outcome**	**Selection of the Reported Result**	**Overall Bias**
Theerth (2018) [57]	Some concerns	Some concerns	Low risk	Low risk	Low risk	High risk
Theerth (2019) [61]	Some concerns	Some concerns	Low risk	Low risk	Low risk	High risk
Tribuddharat (2021) [58]	Low risk	Some concerns	Low risk	Low risk	Low risk	Some concerns
Upton (2017) [5]	Some concerns	Some concerns	Low risk	Low risk	Low risk	High risk
Szental (2015) [8]	Low risk	Some concerns	Low risk	Low risk	Low risk	Some concerns
Dundar (2018) [55]	Some concerns	Some concerns	Low risk	Low risk	Low risk	High risk
Duan (2015) [59]	Some concerns	Some concerns	Low risk	Low risk	Low risk	High risk
Sabourdin (2022) [60]	Some concerns	Some concerns	Low risk	Low risk	Low risk	High risk
Dostalova (2019) [54]	Low risk	Some concerns	Low risk	Low risk	Low risk	Some concerns

QUADAS-2, the Quality of Diagnostic Accuracy Studies-2; RoBANS, Risk of Bias Assessment Tool for Nonrandomized Studies; RoB 2.0, Revised Cochrane risk of bias tool for randomized trials.

**Table 3 jpm-13-01461-t003:** The GRADE evidence quality for each outcome.

Outcome	Number of Studies	Sensitivity		Specificity		Diagnostic Odd Ratio		Quality of Evidence	Certainty of Evidence
		Sensitivity (95% CI)	Hetero- Geneity	Specificity (95% CI)	Hetero- Geneity	Diagnostic Odd Ratio (95% CI)	Hetero- Geneity		
Prediction of intra-operative pain	8 studies	0.81 (0.79–0.83)	I^2^ = 68.2%; Pchi^2^ = 0.043	0.93 (0.92–0.93)	I^2^ = 99.8%; Pchi^2^ < 0.001	2.32 (1.33–3.30)	I^2^ = 61.7%; Pchi^2^ = 0.073	Not serious	⨁◯◯◯ Very low
Prediction of post-operative pain	5 studies	0.90 (0.87–0.93)	I^2^ = 58.7%; Pchi^2^ = 0.064	0.51 (0.49–0.52)	I^2^ = 99.9%; Pchi^2^ < 0.001	3.38 (2.87–3.88)	I^2^ = 81.2%; Pchi^2^ = 0.001	Not serious	⨁◯◯◯ Very low
**Outcome**	**Number of Studies**	**Quality Assessment**					**Hetero-Geneity**	**SMD** **(95% CI)**	**Quality**
		**ROB**	**Inconsistency**	**Indirectness**	**Imprecision**	**Publication Bias**			
Intra-operative opioid use	8 studies	serious	serious	not serious	not serious	NA	I^2^ = 79.2%, P_chi_^2^ < 0.001	−0.262 (−0.450–−0.075)	⨁⨁◯◯ Low

CI; Confidence interval, ROB; Risk of bias, RR; Risk ratio, SMD; Standardized mean difference, NA; Not-applicable.

**Table 4 jpm-13-01461-t004:** Description of studies not included in meta-analysis.

Author (Year)	Main Results			
Theerth (2018) [57]	ANI variable [threshold]		AUC (95CI, SD)	
	ANIi-HR [≤50] ANIm-HR [≤50] ANIi-MBP [≤50] ANIm-MBP [≤50]		0.687 0.672 0.599 0.588	
Jeanne (2014) [44]	ANI variable [threshold]		AUC (95CI, SD)	
	ANI [≤63]		0.92	
	sensitivity = 80%, specificity = 95%, PPV = 94%, and NPV = 79%			
		No stimulation Medain [IQR]	Hemodynamic reactivity Medain [IQR]	*p*-Value
	Patients with hemodynamic reactivity	82.5 [30.3]	47.5 [22.5]	<0.0001
		No stimulation Medain [IQR]	Tibial cut Medain [IQR]	
	Patients without hemodynamic reactivity	80.5 [45]	62 [23]	0.5
Susano (2021) [49]	Remifentanil effect site concentration	ANI before tetanic stimulation Mean ± SD	ANI after tetanic stimulation Mean ± SD	*p*-Value
	0.5 ng/mL	56 ± 16	49 ± 15	0.002
	1.5 ng/mL	68 ± 22	62 ± 22	0.012
	3.0 ng/mL	66 ± 18	59 ± 16	0.009
	5.0 ng/mL	72 ± 16	69 ± 14	0.253
	7.0 ng/mL	71 ± 18	70 ± 16	0.655
Gruenewald (2013) [42]	Remifentanil effect site concentration	ANI before tetanic stimulation Medain [IQR]	ANI after tetanic stimulation Medain [IQR]	*p*-Value
	0 ng/mL 2 ng/mL 4 ng/mL	61 [48–72] 71 [61–88] 88 [70–98]	24 [12–35] 30 [20–40] 13 [5–27]	<0.05 <0.05 <0.05
Coulombe (2021) [41]	Type of stimulus	Prestimulation ANI Median [IQR]	Poststimulation ANI Median [IQR]	*p*-Value
	Intubation Incision	46.8 [39.1–59.2] 73.9 [52.7–87.7]	35.2 [28.6–45.0] 55.7 [45.8–72.1]	<0.0001 0.001
Sabourdin (2013) [51]	ANI was significantly decreased compared with prestimulation values for all remifentanil infusion rates (*p* < 0.05) Stimulation type (tetanic stimulation) Remifentanil infusion rate (0, 2, 0.16, 0.12, 0.08, and 0.04 mcg/kg/min)			
Ledowski (2014) [47]	Type of stimulus	Prestimulation ANI Mean ± SD	Poststimulation ANI Mean ± SD	*p*-Value
	Airway manipulation Skin incision Fentanyl bolus	52.4 ± 19.8 62.7 ± 18.7 53.3 ± 17.9	33.0 ± 11.9 37.9 ± 13.7 59.4 ± 18.7	<0.001 <0.001 <0.05
	The predictive probability value (P_k_) for ANI to predict > 10% changes in HR was 0.61 (SE 0.09) and the P_k_ for >10% changes in SBP 0.59 (SE 0.06).			
Köprülü (2020) [46]	ANI is ineffective in the prediction of potential post-operative pain Pearson correlation: Group I: NRS ≤ 3, r = 0.016 (weak); Group II: NRS 4–6, r = −0.286 (weak); Group III: NRS ≥ 7, r = −0.293 (weak)			
Le Gall (2019) [53]	There was significant reduction of sufentanil use in ANI group compared with non-ANI group (0.15 ± 0.05 vs. 0.17 ± 0.05 mcg/kg/h, *p* = 0.038).			

This table provides a summary of studies that were excluded from the meta-analysis, excluding those that focused on correlation between intra-operative ANI and hemodynamic variables. ANI, analgesia nociception index; ANIi, analgesia nociception index instantaneous; ANIm, analgesia nociception index mean; AUC, area under the receiver operating characteristic curve; IQR, interquartile range; NPV, negative predictive value; PPV, positive predictive value; SD, standard deviation. Data are expressed as mean (95% confidence interval) or Median [IQR].

**Table 5 jpm-13-01461-t005:** Correlation between the intra-operative ANI and hemodynamic variables.

Author (Year)	ANI Variable	Hemodynamic Variables	Correlation Estimate	*p*-Value
Sriganesh (2019) [61]	ANIi	HR MBP	−0.405 −0.415	<0.001 <0.001
Theerth (2019) [56]	ANIi ANIm (2 min average)	ANIi-HR ANIi-MBP ANIm-HR ANIm-MBP	−0.594 −0.534 −0.161 −0.091	<0.001 <0.001 0.007 0.085
Jeanne (2012) [43]	ANI (2 min average)	SBP	−0.348	<0.01
Kommula (2019) [45]	ANI average	HR MAP	−0.280 −0.258	<0.0001 <0.0001

ANI; analgesia nociception index; ANIi, analgesia nociception index instantaneous; ANIm, analgesia nociception index mean; AUC, area under the receiver operating characteristic curve; HR, heart rate; MBP, mean blood pressure; SBP, systolic blood pressure.

## Data Availability

The datasets used and analyzed during the current study are available from the corresponding author upon reasonable request.

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
