# Peer review of "Pain Assessment Using the Analgesia Nociception Index (ANI) in Patients Undergoing General Anesthesia: A Systematic Review and Meta-Analysis"

_jpm, 2023, doi:10.3390/jpm13101461_

Round 1

Reviewer 1 Report

Dear authors,

Thank you for submitting this interesting review article. Please find my comments and address the concerns raised.

There appears to be a discrepancy between the search strategy as mentioned in the methods section and results section. The databases are not the same in both sections. Please make the necessary edits.

The authors have mentioned 3 research questions to describe the inclusion and exclusion criteria. However, this is not the standard way of describing the inclusion and exclusion criteria. Also, there is no direct mention of outcomes (primary and secondary outcomes). It is suggested to describe inclusion and exclusion criteria separately and also mention the outcomes of this review in the methodology.

When mentioning the studies included in the quantitative meta-analysis, mention in brackets how many patients were there in the interventional arm and the control arm.

I suggest mentioning the GRADE evidence (3.7) before the quantitative meta-analysis and then in the description of the meta-analysis, mention the GRADE evidence in the bracket.

Please mention in the methods how the authors graded the heterogeneity: ex-significant, moderate, and low.

The authors have not described/performed a sensitivity analysis. 

Trial sequential analysis of data used for meta-analysis for checking the power and sample sizes of each predefined meta-analytical outcome has not been carried out by authors. The TSA software helps to calculate the required sample size information. I request authors produce the TSA information for primary outcomes.

In the limitations, the authors have mentioned: Fourth, considerable heterogeneity was observed in some outcomes. This is a very important point in the limitations. There is statistical and clinical heterogeneity in the studies included in the analysis. Explain clinical heterogeneity in detail: variable types of surgeries and severity, and medications used were not standardized.

The submission will need technical English language edits, subject to acceptance.

Author Response

We express our heartfelt gratitude for editor and reviewer’s comments.

We revised our manuscript according to the enclosed critique(s) in your comment.

Revision details:

Reviewer #1 comments

  1. There appears to be a discrepancy between the search strategy as mentioned in the methods section and results section. The databases are not the same in both sections. Please make the necessary edits.

Our response: Thank you for reviewer’s comment. It seems like the database remained the same, but there were abbreviation and typo that caused confusion. I have replaced the abbreviation with its full term and also corrected the typo.

Method section: Two independent investigators (KSO and SPL) searched the Ovid-MEDLINE, Ov-id-EMBASE, Cochrane Central Register of Controlled Trials (CENTRAL), and Google Scholar from database establishment to September 2021 using search terms. The search terms were developed in consultation with a medical librarian and included a combina-tion of free text, Medical Subject Headings, and EMTREE terms for the “Analgesia noci-ception index’,” “ANI monitor,” and “surgery.” To obtain comprehensive results, we searched ClinicalTrials.gov and cris.nih.go.kr for ongoing and incomplete clinical trials. We also conducted a search of gray literature using OpenSIGLE (From line 125 to line 132 on page 4)

Result section: A total of 926 studies were identified through searches of Ovid-MEDILINE, Ovid-EMBASE, CENTRAL, Google Scholar, ClinicalTrials.gov, cris.nih.go.kr, and OpenSIGLE (From line 226 to line 227 on page 6)

→ Method section: Two independent investigators (KSO and SPL) searched the Ovid-MEDLINE, Ov-id-EMBASE, Cochrane Central Register of Controlled Trials, and Google Scholar from database establishment to September 2021 using search terms. The search terms were developed in consultation with a medical librarian and included a combina-tion of free text, Medical Subject Headings, and EMTREE terms for the “Analgesia noci-ception index’,” “ANI monitor,” and “surgery.” To obtain comprehensive results, we searched ClinicalTrials.gov and cris.nih.go.kr for ongoing and incomplete clinical trials. We also conducted a search of gray literature using OpenSIGLE (From line 125 to line 132 on page 4)

Result section: A total of 926 studies were identified through searches of Ovid-MEDLINE, Ovid-EMBASE, Cochrane Central Register of Controlled Trials, Google Scholar, ClinicalTrials.gov, cris.nih.go.kr, and OpenSIGLE (From line 226 to line 227 on page 6)

  1. The authors have mentioned 3 research questions to describe the inclusion and exclusion criteria. However, this is not the standard way of describing the inclusion and exclusion criteria. Also, there is no direct mention of outcomes (primary and secondary outcomes). It is suggested to describe inclusion and exclusion criteria separately and also mention the outcomes of this review in the methodology.

Our response: Thank you for reviewer’s comment. The research topics corresponding to Q1 serve as the primary outcome, while Q2 and Q3 are considered secondary outcomes for additional analysis. Consequently, the inclusion and exclusion criteria are the same for all three topics. As suggested by the reviewer, to avoid confusion, we have changed the terminology from Q1, Q2, and Q3 to "Primary Outcome," "Secondary Outcome #1," and "Secondary Outcome #2." Thank you for the valuable feedback.(From line 87 to line 110 on page 3)

  1. When mentioning the studies included in the quantitative meta-analysis, mention in brackets how many patients were there in the interventional arm and the control arm.

Our response: Thank you for editor’s comment. According to editor’s comment, we changed the manuscript. (From line 295 to line 400 on page 14-20).

  1. I suggest mentioning the GRADE evidence (3.7) before the quantitative meta-analysis and then in the description of the meta-analysis, mention the GRADE evidence in the bracket.

Our response: Thank you for reviewer’s comment. As the reviewer's suggestion, we have reordered the GRADE section to 3.4 and adjusted the table numbers accordingly. Additionally, we have included the GRADE within the bracket. (From line 274 to line 400 on page 14-20).

  1. Please mention in the methods how the authors graded the heterogeneity: ex-significant, moderate, and low.

Our response: Thank you for reviewer’s comment. We added information about the GRADE assessment to the section where the existing method for evaluating heterogeneity is described (From line 210 to line 212 on page 6).

  1. The authors have not described/performed a sensitivity analysis.

Our response: Thank you for editor’s comment. We have conducted a sensitivity analysis, and there was no significant difference.

(From line 210 to line 212 on page 6).

  1. Trial sequential analysis of data used for meta-analysis for checking the power and sample sizes of each predefined meta-analytical outcome has not been carried out by authors. The TSA software helps to  calculate  the  required  sample  size    I  request  authors  produce  the  TSA information for primary outcomes.

Our response: Thank you for reviewer’s considerate comment. I agree with your comment that The TSA analysis helps to check the power and sample size calculate  the  required  sample  size  information. TSA analysis adopts sequential analysis methods for systematic reviews and meta-analyses, but TSA is different from sequential analysis in a single trial in that the enrolled unit is not a patient but a study. However, TSA program does not provide tools for diagnostic study and standardized mean difference for continuous variables currently. As intraoperative opioid use was compared for different type of opioid, we cannot use mean difference for this outcome. Therefore, to my knowledge, we do not have methods for applying TSA for this manuscript. (Hyun Kang. Trial sequential analysis: novel approach for meta-analysis. Anesthesia and Pain Medicine 2021;16(2):138-150.)

  1. In the limitations, the authors have mentioned: Fourth, considerable heterogeneity was observed in some outcomes. This is a very important point in the limitations. There is statistical and clinical heterogeneity in the studies included in the analysis. Explain clinical heterogeneity in detail: variable types of surgeries and severity, and medications used were not standardized.

Our response: Thank you for reviewer’s comment. As the reviewer's suggestion, we incorporated the description. (From line 502 to line 505 on page 23)

Reviewer 2 Report

Well written but a bit too long. This review is a good starting point for readers who want a comprehensive insight into intraoperative nociception monitoring.

While I think that comparison with qNox would be interesting, it would make the article even longer so it's omission is not a major problem.

Author Response

We express our heartfelt gratitude for editor and reviewer’s comments.

We revised our manuscript according to the enclosed critique(s) in your comment.

Revision details:

Reviewer #2 comments

  1. Well written but a bit too long. This review is a good starting point for readers who want a comprehensive insight into intraoperative nociception monitoring.

While I think that comparison with qNox would be interesting, it would make the article even longer so it's omission is not a major problem.

Our response: We are grateful for your positive feedback and suggestions. Regarding your comment about comparing our study with qNox, we acknowledge the value of such a comparison. However, considering the potential lengthening of the article and the scope of our current work, we have decided not to include this comparison.

Once again, we thank you for your valuable input.

Reviewer 3 Report

Respected authors,

I enjoyed reading your article and I suggested that the editor accept it for publication in its original form.

the article you sent me for review meets high professional standards, and the results it describes are a useful contribution to clinical practice in anesthesiology and in the treatment of postoperative pain. I suggest that the article be published in its present form.

Author Response

We express our heartfelt gratitude for editor and reviewer’s comments.

We revised our manuscript according to the enclosed critique(s) in your comment.

Revision details:

Reviewer #3 comments

  1. I enjoyed reading your article and I suggested that the editor accept it for publication in its original form.

the article you sent me for review meets high professional standards, and the results it describes are a useful contribution to clinical practice in anesthesiology and in the treatment of postoperative pain. I suggest that the article be published in its present form.

Our response: Thank you very much for your thoughtful and encouraging review of our article. Your kind words motivate us to continue our research efforts in this area, and we are thrilled that you found our article to meet high professional standards and deem it a valuable contribution to clinical practice. We are grateful for your positive feedback and suggestions.

Round 2

Reviewer 1 Report

Dear authors,

Thank you for responding to the comments and also making changes based on the comments.

Technical and grammatical edits will be required, if accepted, during proof reading stage.